# Stabilization of Hypoxia-Inducible Factors and BNIP3 Promoter Methylation Contribute to Acquired Sorafenib Resistance in Human Hepatocarcinoma Cells

**DOI:** 10.3390/cancers11121984

**Published:** 2019-12-09

**Authors:** Carolina Méndez-Blanco, Flavia Fondevila, Paula Fernández-Palanca, Andrés García-Palomo, Jos van Pelt, Chris Verslype, Javier González-Gallego, José L. Mauriz

**Affiliations:** 1Institute of Biomedicine (IBIOMED), University of León, Campus of Vegazana s/n, 24071 León, Spain; cmenb@unileon.es (C.M.-B.); ffonp@unileon.es (F.F.); pferp@unileon.es (P.F.-P.); jgonga@unileon.es (J.G.-G.); 2Centro de Investigación Biomédica en Red de Enfermedades Hepáticas y Digestivas (CIBERehd), Instituto de Salud Carlos III, Av. de Monforte de Lemos, 5, 28029 Madrid, Spain; 3Service of Oncology, Complejo Asistencial Universitario de León, Calle Altos de Nava, s/n, 24001 León, Spain; dfijmg@unileon.es; 4Laboratory of Clinical Digestive Oncology, Department of Oncology, KU Leuven and University Hospitals Leuven and Leuven Cancer Institute (LKI), 3000 Leuven, Belgium; jos.vanpelt@kuleuven.be (J.v.P.);

**Keywords:** BNIP3, hepatocarcinoma, HIF, hypoxia, resistance, sorafenib

## Abstract

Despite sorafenib effectiveness against advanced hepatocarcinoma (HCC), long-term exposure to antiangiogenic drugs leads to hypoxic microenvironment, a key contributor to chemoresistance acquisition. We aimed to study the role of hypoxia in the development of sorafenib resistance in a human HCC in vitro model employing the HCC line HepG2 and two variants with acquired sorafenib resistance, HepG2S1 and HepG2S3, and CoCl_2_ as hypoximimetic. Resistant cells exhibited a faster proliferative rate and hypoxia adaptive mechanisms, linked to the increased protein levels and nuclear translocation of hypoxia-inducible factors (HIFs). HIF-1α and HIF-2α overexpression was detected even under normoxia through a deregulation of its degradation mechanisms. Proapoptotic markers expression and subG1 population decreased significantly in HepG2S1 and HepG2S3, suggesting evasion of sorafenib-mediated cell death. HIF-1α and HIF-2α knockdown diminished resistant cells viability, relating HIFs overexpression with its prosurvival ability. Additionally, epigenetic silencing of Bcl-2 interacting protein 3 (BNIP3) was observed in sorafenib resistant cells under hypoxia. Demethylation of BNIP3 promoter, but not histone acetylation, restored BNIP3 expression, driving resistant cells’ death. Altogether, our results highlight the involvement of HIFs overexpression and BNIP3 methylation-dependent knockdown in the development of sorafenib resistance in HCC. Targeting both prosurvival mechanisms could overcome chemoresistance and improve future therapeutic approaches.

## 1. Introduction

Hepatocarcinoma (HCC), a common malignancy worldwide with a quick rise in incidence, is one of the main causes of cancer-related death because of its high mortality rate [1]. Most of HCC cases are diagnosed in advanced stages, when there are no healing therapies available and the first-line therapy employed is the palliative treatment with sorafenib (BAY 43-9006, Nexavar^®^), an oral multikinase inhibitor with antiproliferative, proapoptotic, and antiangiogenic properties [2]. The antiproliferative activity responds to the suppression of the serine/threonine kinases Raf-1 and B-Raf, with the consequent inhibition of the mitogen-activated protein kinases (MAPK)/extracellular signal-regulated kinases (ERK) signaling pathway. The proapoptotic action of the drug is associated with the inhibition of eIF4E phosphorylation and subsequent downregulation of the antiapoptotic factor induced myeloid leukemia cell differentiation protein (Mcl-1) translation [1,3]. Furthermore, sorafenib exerts its antiangiogenic activity by targeting the tyrosine kinase receptors mast/stem cell growth factor receptor (c-Kit), FMS-like tyrosine kinase (FLT-3), vascular endothelial growth factor receptors 2 and 3 (VEGFR-2, VEGFR-3), and platelet-derived growth factor receptor (PDGFR-β) [3].

The results obtained from the sorafenib hepatocellular carcinoma assessment randomized protocol (SHARP) clinical trial showed that this drug is effective and safe, encouraging the approval of sorafenib for advanced HCC [1]. Nonetheless, the efficacy of sorafenib is quite brief, since it extends the patients survival only in a few months due to the appearance of resistant HCC cells [2]. While some patients present primary resistance to sorafenib because of the intrinsic genetic heterogeneity of HCCs, in most cases long-term exposure to the drug promotes the acquired resistance to sorafenib [4]. The interplay between the phosphatidylinositol-3-kinase (PI3K)/protein kinase B (Akt) and janus tyrosine kinase (JAK)/signal transducer and activator of transcription (STAT) pathways, disabling apoptotic signals, dysregulation of cell cycle control, epigenetic regulation, epithelial–mesenchymal transition and the hypoxia-inducible response are some of the mechanisms involved in the cells sensitivity to sorafenib decline [2].

Hypoxia is a shared feature of solid tumors, like HCC, which originates by oxygen supply default as a result of an insufficient vascularization [5]. Tumor cells can activate prosurvival adaptive machinery against this unfavorable cellular situation favoring disease progression. Hypoxia-inducible factors (HIFs) are the main mediators in cellular adaptation to hypoxic microenvironment, regulating a broad diversity of genes implicated in tumor survival [6]. Overexpression of HIF-1α and HIF-2α has been noticed in several liver diseases, including HCC [7]. These transcription factors are composed by a α subunit, whose levels depend on the balance between its oxygen-dependent degradation and its oxygen-independent synthesis, and by a constitutively expressed β subunit [6,8]. HIF-α is constituently degraded under normal oxygen supply through the proline residues hydroxylation by the prolyl hydroxylases (PHDs), using oxygen as cofactor, which allows the interaction of HIF-α and the von Hippel-Lindau (VHL) tumor suppressor. The ubiquitin E3 ligase protein recognizes VHL, promoting the HIF-α ubiquitination and its proteasomal degradation [8,9]. Conversely, under hypoxia conditions, PHDs activity is restricted by oxygen inaccessibility, leading to the stabilization of HIF-α subunits that allows its protein accumulation and its translocation to the nucleus. There, HIF-α heterodimerizes with HIF-β subunit and binds to the hypoxia-response elements (HREs) in the promoters of its targets genes, which are involved in tumor progression and therapy resistance, driving its transcription [5,8].

Antiangiogenic activity of sorafenib derives from the blockage of HIF-1α/VEGF pathway [2,4]. Although this property is able to avoid HCC progression in the early stages, decreased microvessel density resulting from the antiangiogenic action of sustained sorafenib administration enhances intratumoral hypoxia, leading to HIFs overexpression [4]. Therefore, the hypoxic microenvironment tightly links to the selection of HCC cell clones adapted to oxygen and nutrient deficiency, and to the acquisition of sorafenib resistance [2,4].

Bcl-2 interacting protein 3 (BNIP3) is another hypoxia-regulated protein belonging to the BH3-only Bcl-2 family that has proven to be involved in apoptosis, necrosis, autophagy, and mitophagy under hypoxia or ischemia [10,11]. BNIP3, which may be activated by HIF-1α due to its HREs-containing promoter, is expressed in hypoxic tumoral areas, contributing to hypoxia-induced cell death [12]. In fact, loss of BNIP3 expression correlates with poorer survival of patients and chemoresistance in different types of cancer and is broadly subjected to epigenetic alterations, highlighting histone deacetylation and DNA methylation [10,12,13,14,15,16,17,18].

In the present research, we employed a well-established in vitro model of HCC with acquired resistance to sorafenib to investigate more accurately the underlying hypoxia-related mechanisms involved in the chemoresistance process, focusing on the role of HIFs and BNIP3. Here, we reported that HIFs stabilization and subsequent overexpression, in addition to BNIP3 promoter methylation-dependent silencing, lead to evasion of sorafenib-mediated apoptosis, contributing to drug resistance acquisition in HCC cells. These results satisfy the required demand to set the bases for the posterior translational investigation in this field.

## 2. Results

### 2.1. Resistant Hepatocarcinoma (HCC) Cell Lines Show a More Aggressive Growth than the HepG2 Parental Line under Both Normoxia and Hypoxia

Sorafenib has been shown to be effective against advanced HCC; however, tumor cells are capable of developing resistance mechanisms after prolonged treatment, thus evading its antitumor effects. Hypoxia, a characteristic feature of solid tumors, is one of the main contributors to the acquisition of drug resistance, including to sorafenib in HCC, and it is closely related to a worse prognosis [7]. In order to study this resistance phenomenon and the hypoxia implication on it, we decide to work with the human HCC cell line HepG2 and two derived sorafenib resistant cell lines, defined as HepG2S1 and HepG2S3, which are always cultured with 6 μM sorafenib to maintain drug resistance [19].

We first evaluated the growth dynamic of the two resistant cell lines against the parental line in absence and presence of 6 μM sorafenib along five days, both under normoxia and hypoxia conditions. Sorafenib treatment inhibited the cellular growth of the HepG2 parental line under both conditions. However, HepG2S1 and HepG2S3 cells showed a raised growth capacity under normoxia and hypoxia, which was even higher than that observed in parental line in sorafenib absence. Among resistant cell lines, HepG2S1 exhibited the highest growth, being superior to that observed in HepG2S3 after day 1 in normoxia and day 3 in hypoxia. Furthermore, the hypoxic microenvironment was able to enhance the difference in cell growth observed between both resistant lines and the parental line without sorafenib (Figure 1a). These data agree with those derived from the study of the proliferative capability by immunofluorescence analysis of Ki67 (Figure 1c), suggesting altogether that resistant cell lines present a more aggressive phenotype.

Intratumoral hypoxia has been related to the development of sorafenib resistance in HCC [7]. Therefore, in addition to contrast the growth between the different cell lines, we compared the growth of each line by separately between normoxia and hypoxia (Figure 1b). We observed that growth of HepG2 parental cells without treatment was reduced by inducing hypoxia, while the two resistant lines maintained a similar growth rate under both oxygen situations (Figure 1b), indicating that resistant cells might have active adaptive mechanisms related to hypoxic response.

### 2.2. Sorafenib Resistant Cell Lines Overexpress Hypoxia-Inducible Factors (HIFs) and Display a Deregulation in the HIF-1α Degradation Mechanisms

Hypoxic environment supposes a cellular stress that promotes an adaptive response through the stabilization of HIFs. HIF-1α is the main factor that regulates cellular response to hypoxia, being involved in tumor cells adaptation to intratumoral hypoxia, as well as in acquisition of resistance to chemotherapeutic drugs such as sorafenib. HIF-2α factor also participates in HCC cells’ response to lack of oxygen supply and could be involved in the evasion of antitumor signals of sorafenib by liver tumor cells [2,20,21]. Considering the indisputable participation of hypoxia in the development of chemoresistance, we decided to study how HepG2S1 and HepG2S3 resistant cells respond against hypoxia induction by analyzing HIF-1α and HIF-2α expression along 48 h.

The HepG2 parental line showed a progressive increase in HIF-1α protein expression after hypoxia induction, whereas sorafenib addition prevented its accumulation. HepG2S3 resistant cells exhibited higher HIF-1α expression than the HepG2 line treated with sorafenib, appreciating similar levels than those registered for the parental line without exposure to the drug. Nevertheless, it was the HepG2S1 resistant line in which we observed the greatest HIF-1α overexpression. In the case of HIF-2α, both resistant cell lines showed an increase in its protein expression in relation to the parental HepG2 line with/without sorafenib, where no detectable levels were observed. As loading control, we initially used β-actin; however, because of its expression there was no constant between the different analyzed cell lines, so we employed the proliferation cell nuclear antigen (PCNA) (Figure 2a). Such HIFs expression patterns were confirmed through expression analysis of both hypoxic markers by immunofluorescence and confocal microscopy (Figure 2b). Moreover, nuclear translocation of HIF-1α and HIF-2α was assessed, showing a higher translocation of both transcription factors in the HepG2S1 and HepG2S3 resistant cell lines than HepG2 cells with or without treatment (Figure 2b).

It should be mentioned that even under normoxic conditions, resistant cells exhibited a significant increment in the expression of HIF-1α, much more remarkable in the HepG2S1 line, and of HIF-2α (Figure 2a). Given the alteration observed in HIF-1α expression pattern of sorafenib resistant cells, the main factor that participates in adaptive response to hypoxia, we decided to study the synthesis and degradation processes that HIF-1α undergoes. Parental cells in absence of sorafenib and HepG2S1 and HepG2S3 cells were incubated in normoxia and hypoxia during 24 h, and individually treated with chemical inhibitors of protein synthesis (cycloheximide (CHX)) and degradation (MG132) along the last 6 h. As shown in Figure 2c, blockage of degradation with MG132 caused a strongly accumulation of HIF-1α in all cell lines analyzed. On the other hand, whereas protein synthesis inhibition with CHX resulted in a significant reduction of HIF-1α levels in HepG2 cell line, HIF-1α expression was not affected in sorafenib resistant lines (Figure 2c). These results suggest that the HIF-1α degradation process seems to be abrogated in the HepG2S1 and HepG2S3 cells.

### 2.3. Resistant Cells Can Evade Sorafenib-Mediated Cell Death, Being HIFs Involved in This Lack of Cell Sensitivity to Sorafenib

Cell death processes, among which apoptosis is located, are usually altered in chemoresistant tumor cells [5]. For this reason, we wanted to check if our in vitro HCC model of sorafenib resistance has some disturbance in cell death compared to the parental model.

Microarray analysis was performed to assess the differences in gene expression between the HepG2 parental line and the resistant line HepG2S1. KEGG-pathway for apoptosis (hsa04210) was found to be significantly enriched. We found a significant suppression of the proapoptotic genes BAX and TNFRSF10B in HepG2S1 cells, while in the prosurvival group of genes, we saw a 33-fold upregulation of BIRC3, 4-fold increase of PTPN13, and doubling of BCL2. Moreover, several caspases showed a differential expression. It outstands the CASP3 suppression in HepG2S1 to about 25% level compared to HepG2, although CASP8 and CASP10 were also significantly reduced. To be noted, as the HepG2S1 are adapted to sorafenib over a period of several months, other compensatory processes have also been induced, some genes are shifted in the opposite direction; furthermore, the gene expression data suggest a suppression of apoptosis (Table 1).

Cell cycle evaluation showed a drastic increase in the number of HepG2 cells in subG1 phase when sorafenib was added, achieving around 10% dead cells vs. the 3% observed in these cells without treatment (Figure 3a). Percentage of subG1 cells was similar between HepG2S1 and HepG2S3 resistant lines (~5.5%), but significantly lower with respect to that detected in sorafenib-treated parental line (Figure 3a). As seen in Figure 3b, treatment of HepG2 line with 6 µM sorafenib under hypoxia during 24 and 48 h increased the expression of both apoptosis markers in comparison to non-treated parental cells. In contrast, protein levels of Bax and cleaved caspase-3 were notably reduced in the two resistant cell lines (Figure 3b), suggesting altogether that HepG2S1 and HepG2S3 cells have developed some mechanisms to evade cell death signals.

Additionally, the interesting findings observed in HIFs expression and in apoptosis process in both HepG2S1 and HepG2S3 sorafenib resistant cells, which probably are interrelated, made us assess the implication of HIFs in survival ability. To this, we silenced both HIF-1α and HIF-2α using small interfering RNAs (siRNAs) and tested its impact on cell viability at 24 h post-hypoxia induction. HIF-1α knockdown was able to reduce it in a significant way in both resistant cell lines, especially in HepG2S1 cells, where it approximately declined the viability in a 50%. In addition, HIF-2α downregulation triggered cell death in both resistant cell lines, being more pronounced in HepG2S3 cells with respect to HepG2S1 (~75% vs. ~40% cell viability, respectively) (Figure 3c). These data suggest that HIFs play an important role in sorafenib resistant cells’ survival.

### 2.4. Methylation-Dependent Downregulation of Bcl-2 Interacting Protein 3 (BNIP3) Participates in Hypoxia-Mediated Sorafenib Resistance

BNIP3 is a proapoptotic protein of the Bcl-2 family regulated by hypoxic conditions [11]. In view of the implication of hypoxia in our sorafenib resistant model, we compared BNIP3 expression between the different cell lines along 48 h under hypoxic conditions. BNIP3 protein levels were reduced in HepG2 cells when sorafenib was added, finding a more pronounced decrease in the resistant cells HepG2S1 and HepG2S3 (Figure 4a,b). Likewise, BNIP3 mRNA levels showed the same trend, which indicates that BNIP3 downregulation in resistant cells was due to an upstream process (Figure 4c).

Epigenetic modifications have been associated with gene silencing. Many tumor suppressor genes are known to be disabled by epigenetic alterations that include DNA methylation of its 5′ regions and histone deacetylation [11]. First, we evaluated histone deacetylation through the employment of 10, 50, and 100 nM of the histone deacetylases (HDACs) inhibitor trichostatin-A (TSA). No changes were detected on BNIP3 expression in any of the resistant cell lines after 24, 48, and 72 h of exposure to the different doses of TSA, discounting histone deacetylation as responsible of BNIP3 silencing under hypoxia (Figure 4d). Thus, we examined the methylation status of the BNIP3 promoter, obtaining raised levels of methylated DNA in HepG2S1 and HepG2S3 than HepG2 cells (Figure 4e). The DNA methyltransferase (DNMT) inhibitor 5-aza-2′-deoxycytidine (5-Aza) has been demonstrated to be effective in preventing promoter methylation from the lowest dose tested (10 μM) (Figure 4f) and reestablishing hypoxia-induced expression of BNIP3 at both mRNA (Figure 4g) and protein (Figure 4h) levels. To elucidate that recovered BNIP3 expression is responsible for cell death induction in HepG2S1 and HepG2S3 cells, we silenced BNIP3 and simultaneously treated with 5-Aza under hypoxia. In both resistant cell lines, demethylation by 5-Aza significantly promoted cell death, while BNIP3 knockdown triggered cell viability of sorafenib resistant cell lines, observing a diminution on the cell death percentage with respect to 5-Aza-treated control siRNA cells (Figure 4i). This suggests that BNIP3 epigenetic downregulation likely plays a key role in the sorafenib resistance and may be a useful molecular target for HCC therapy.

## 3. Discussion

Initially, we characterized the resistant phenotype by analyzing the cell growth dynamics and Ki67 expression of both sorafenib resistant lines, HepG2S1 and HepG2S3, against the HepG2 parental line in the absence and presence of sorafenib, under conditions of normoxia and hypoxia. Treatment with sorafenib caused a decrease in the proliferative capacity of the parental line HepG2 under both oxygen conditions. This outcome was previously observed under normoxia with the human HCC lines HepG2, Hep3B, and Huh7 [22,23], and in Hep3B cells under both oxygen conditions [24]. In the present research, hypoxia suppressed the cellular growth of HepG2 line in comparison with normoxia, since it triggered a cellular stress. Contrarily, a study with the HCC line Hep3B found that hypoxia does not decrease cell viability [24]. While hypoxia was induced for 48 h [24], here we appreciated relevant differences in HepG2 cell growth between normoxia and hypoxia from day 2, reaching the maximum on day 5. Hence, hypoxia effects on dynamic growth might depend on exposure time to hypoxia as well as the HCC cell line tested.

When we assessed the growth and Ki67 expression of sorafenib resistant cells HepG2S1 and HepG2S3, a higher capacity of proliferation was observed with respect to HepG2 parental line treated with sorafenib and even against parental line without treatment. This fast growth in both resistant lines may correspond to its greater invasive potential already described [19]. Such a difference in the growth rate between the lines resistant to sorafenib and the HepG2 parental line was potentiated under hypoxic conditions. Unlike what happened with HepG2 cells, no differences were detected in the growth of the HepG2S1 and HepG2S3 lines between normoxia and hypoxia, which suggest that resistant cells display adaptive mechanisms related to hypoxic conditions. Hypoxia contribution to chemoresistance development in HCC has been described after the administration of drugs, such as 5-fluorouracil (5-FU) [25], doxorubicin, and cisplatin [26]. Involvement of hypoxic environment in the acquired resistance to sorafenib has also been reported in renal cancer [27,28], gastric cancer [29], and acute myeloid leukaemia [30]. In samples of patients with untreated HCC sensitive and resistant to sorafenib, intratumoral hypoxia increased in patients resistant to sorafenib as opposed to sensitive ones [20].

Sorafenib treatment of the HepG2 parental line reduced the protein levels and nuclear translocation of HIF-1α under hypoxia, while almost no expression of HIF-2α was detected either in the absence or in the presence of sorafenib. It has already been seen that sorafenib decreases HIF-1α levels in HCC cell lines [31,32] by inhibiting its protein synthesis [31]. We observed a noteworthy increase in both HIF-1α protein expression and nuclear translocation in both resistant lines HepG2S1 and HepG2S3 compared to the sorafenib-treated parental line HepG2, highlighting the outstanding overexpression of this factor in HepG2S1 line with respect to non-treated HepG2 one. A higher HIF-1α expression has been reported in HCC samples from sorafenib resistant patients with respect to sorafenib-sensitive or non-treated patients [20]. Likewise, in both in vitro and in vivo models, prolonged sorafenib treatment of liver tumor cells lead to an increased HIF-1α expression [20,33,34]. Interestingly, HIF-2α levels were upregulated in both resistant lines against the parental line in the absence and presence of sorafenib, finding also a significant higher HIF-2α translocation to the nucleus. Previous investigations in HCC in vitro models showed that HIF-2α overexpression promotes sorafenib resistance [21], whilst HIF-2α downregulation enhances the antitumor actions of sorafenib [35]. Therefore, the increase in the expression and nuclear translocation of HIF factors in resistant cells seems to be firmly involved in the acquisition of sorafenib resistance and associated with its more aggressive growth. Similar results were reported in various human ovarian cancer lines undergoing stem cell-like properties, where HIF-2α upregulation mediated adriamycin resistance [36].

In the present study, HepG2S1 and HepG2S3 lines overexpressed both HIFs even under normoxia. This interesting result could be the consequence of the development of alterations in the synthesis and/or degradation mechanisms of HIFs during sorafenib resistance acquirement. However, due to HIF-2α expression being practically undetectable in HepG2 parental cells, our control cell line, we could only evaluate the potential alteration of such mechanisms by assessing HIF-1α expression. Although protein synthesis was not affected in any cell lines analyzed, inhibition of this process decreased significantly HIF-1α expression in HepG2 line but not modified HIF-1α levels in HepG2S1 and HepG2S3 cells. This suggests the abolishment of the degradation mechanisms of HIF-1α in the sorafenib resistant lines that may connect to its capacity of survival. Accordingly, HIF-1α stabilization by preventing its proteasomal degradation has been associated with the acquisition of sorafenib or doxorubicin chemoresistance in HCC [37,38]. In this way, promotion of VHL-dependent HIF-1α degradation contributes to surpass sorafenib resistance [20].

In addition to hypoxia, other mechanisms, such as dysregulation of cell cycle control and evasion of cell death, are a hallmark in the tumor progression and in the development of cancer drug resistance [5]. In fact, hypoxia itself can cause alterations in these processes that regulate tumor cells’ survival [7]. The percentage of HepG2 cells in the subG1 phase experienced a dramatic increase when adding sorafenib. These results agree with the sorafenib effect on the cell cycle distribution of different HCC cell lines, including HepG2 [39,40]. However, the percentage of subG1 HepG2S1 and HepG2S3 cells was significantly lower than that of the parental line also treated with the drug, suggesting a reduced cell death rate in the resistant cells. This result was also described in MCF-7 tamoxifen-resistant cells [41].

Furthermore, sorafenib administration increased Bax and cleaved caspase-3 protein levels in the parental line HepG2. Similar results were obtained in other studies where sorafenib treatment led to an increase in proapoptotic signals [23,39,42,43]. On the contrary, protein levels of Bax and cleaved caspase-3 were considerably reduced in the resistant lines HepG2S1 and HepG2S3, in agreement with the results from the relative RNA expression for apoptosis markers determined by microarray analysis. A research accomplished with two sorafenib resistant HCC lines demonstrated that both overexpressed the antiapoptotic protein Bcl-2, leading to a reduced apoptosis induction [44]. Moreover, viability of HCC cells with miR-34a overexpression, a Bcl-2 inhibitor, was suppressed in comparison to those with normal miR-34a expression [45]. Implication of the antiapoptotic protein Bcl-xl in sorafenib resistance has been also reported in both in vitro and in vivo studies [46,47]. Liang et al. [20] found a lower apoptotic index in sorafenib resistant samples from HCC patients than in sensitive ones. Decreased expression of both apoptotic markers in our sorafenib resistant cells occurred parallel to the HIFs expression increase. Taken together, these results indicate that HepG2S1 and HepG2S3 cell lines have developed mechanisms for evading apoptosis as a survival strategy against sorafenib.

To determine if HIFs overexpression is involved in the survival capacity of the resistant tumor cells, knockdown of HIF-1α and HIF-2α was carried out. Silencing of HIF-1α in resistant cells promoted the decrease of viability, even reducing it in half in the case of HepG2S1 cells. Besides, HIF-2α downregulation induced cell death in both resistant cell lines. These data stand out as the involvement of HIFs in the survival ability of sorafenib resistant cells, suggesting that targeting these factors could overcome sorafenib resistance.

Accordingly, sorafenib combination with different compounds, such as EF24 or genistein, overcomes hypoxia-mediated sorafenib resistance in HCC through HIF-1α downregulation [20,48]. HIF-1α silencing is also effective to decrease invasive potential and rise chemosensitivity of human glioblastoma (LN229) and astrocytoma (U-251MG) cell lines to temozolomide [49], and to potentiate sorafenib-mediated apoptosis in HCC cells [50]. Furthermore, microRNA overexpression stimulates apoptotic cell death by inhibiting HIF-1α both in vitro and in vivo HCC models [33,51], among other tumors [52]. Other researchers have focused on the HIF-2α contribution to sorafenib resistance in HCC. Zhao et al. [21] described that HIF-2α siRNA transfection downregulates TGF-α/EGFR pathway, promoting apoptosis in HCC models, while He et al. [36] observed that adriamycin-resistant ovarian cancer cells can be significantly sensitized by targeting breast cancer resistance protein (BCRP) expression. Patient-derived colon cancer cells in advanced clinical stages exhibiting drug resistance surpassed 5-FU or CCI-779 resistance when combined with HIF-2α-specific inhibition [53]. The addition of metformin to sorafenib treatment inhibits HIF-2α protein expression in vitro and in an orthotopic xenograft mouse model, which allows recovery of HCC cells sensitivity to sorafenib-induced apoptosis [54]. Therefore, targeting HIF-1α and HIF-2α together could be an interesting approach. Ma et al. [32] reported that 2-methoxyestradiol repressed nuclear translocation and expression of HIF-1α and HIF-2α, synergizing with sorafenib to stop tumoral proliferation and induce apoptosis of HCC cells. Sodium orthovanadate administration is also able to reduce the expression and nuclear translocation of both factors, overcoming sorafenib resistance in HCC cells [55].

Given the key role displayed by hypoxia response in the development of resistance to sorafenib in our model, we explored the possible implication of BNIP3, whose expression is also regulated by the hypoxic microenvironment. BNIP3 is a mitochondrial hypoxia responsive protein involved in the maintenance of cellular homeostasis during hypoxia or ischemia through regulation of apoptosis, necrosis, autophagy, and mitophagy [10,11]. BNIP3 possesses promoter containing HREs; thus, its expression may be activated by HIF-1α [12]. Nonetheless, other transcription factors apart from HIF-1α, such as pleomorphic adenomas gene like2 (PLAGL2), Forkhead box O3 (FOXO3), E2F1, and nuclear factor κB (NF-κB) have also been described as regulators of BNIP3 transcription [11].

BNIP3 downregulation has been linked to poorer patient survival and cell proliferation in pancreatic, colorectal, renal, and HCC tumors [10,12,13,14,15,16,17,18,56,57,58,59]. Erkan et al. [14] found lower BNIP3 mRNA levels in an in vitro model and in four out of eight pancreatic cancer cell lines. Moreover, scientific evidence has confirmed a close association between BNIP3 silencing and chemoresistance acquisition to 5-FU [10,17] and oxaliplatin [16] in colorectal cancer, and to gemcitabine and 5-FU in pancreatic ductal adenocarcinoma [14]. In the present study, lower protein and mRNA levels of BNIP3 were detected in resistant cells compared to parental ones. These data indicate that BNIP3 expression absence in sorafenib resistant cells could be due to an upstream process alteration.

Several genes are recognized to be silenced by epigenetic modifications, promoting pathology-related physiological functions in the cells [11]. Histone deacetylation influences chromatin tension and, consequently, the regulation of the transcription process, inhibiting or silencing genes like tumor suppressors [12]. To assess the potential role of histone deacetylation on BNIP3 downregulation in our sorafenib-resistant HCC model, we examined the effect of the HDAC inhibitor TSA. We found no changes after TSA treatment, rejecting histone deacetylation as accountable for BNIP3 suppression. Thereupon, we studied the promoter methylation as a possible mechanism of BNIP3 expression abrogation. Hypermethylation occurs at promoter CpG islands located in the gene promoters and contributes to the functional inactivation of tumor suppressor genes, with DNMTs being responsible for DNA methylation [11,12]. Here, we reported a great rate of BNIP3 promoter methylation in both resistant cells. Furthermore, BNIP3 expression under hypoxia was reestablished in sorafenib resistant cells after 5-Aza administration, which resulted in reduced resistant cells viability; while BNIP3 silencing led to a higher survival of resistant cells subjected to 5-Aza treatment. Hence, chemoresistance to sorafenib is likely related to aberrant methylation and the subsequent epigenetic silencing of BNIP3, suggesting BNIP3 upregulation could prompt chemosensitivity of HCC cells.

In accordance with our results, several studies have related BNIP3 silencing with methylation of its promoter as a protective mechanism of cell death in leukemia, pancreatic, and colorectal tumors [10,13,15,18,57,58,60,61], resulting from DNMT1 activity by the mitogen-activated protein kinase in pancreatic cancer [13] and from DNMT1/DNMT3B in colorectal cancer [10]. In most cases, 5-Aza was used to restore normal BNIP3 expression, sensitizing pancreatic cancer cells via hypoxia-mediated apoptosis promotion [13,15,18] and busulfan-resistant myeloid leukemia cells by upregulation of proapoptotic proteins, including BNIP3 [60]. This phenomenon has been largely studied in colorectal cancer, where 5-Aza recovered BNIP3 expression as a biosensitizer pretreatment of irinotecan [61] and to increase chemosensitivity to 5-FU in colorectal cancer [10]. Additionally, BNIP3 upregulation in a DNA demethylation-dependent manner in colorectal cancer has been reported to overcome apoptosis resistance after verticillin A treatment [58] and related to DNMT1 inhibition following radiotherapy and chemotherapy [57].

Nevertheless, Shao et al. [12] reported that histone deacetylation, but not methylation, represents the primary cause of BNIP3 inactivation in renal cell carcinoma; and the acetylation status is restored with TSA, leading to cell growth blockage and apoptosis induction. Besides, other authors have evidenced knockdown of BNIP3 in colorectal cancer cells by both methylation of its 5′ CpG island and deacetylation of histone in that region, showing that 5-Aza and TSA administration recovered BNIP3 expression [56,59], even better in combination [59].

## 4. Materials and Methods

### 4.1. Cell Culture

The human HCC cell line HepG2 was obtained from the American type culture collection (Manassas, VA, USA) and the two variants of this cell lines which undergo resistance to sorafenib (HepG2S1 and HepG2S3) were generated by the laboratory of hepatology of the University Hospitals Leuven [19]. Cells were grown in Dulbecco’s modified eagle’s medium (DMEM)-high glucose, supplemented with 10% fetal bovine serum and penicillin/streptomycin (100 U/mL), and they were cultured under a humidified 5% CO_2_ atmosphere at 37 °C. Cell culture reagents were purchased from Sigma-Aldrich (St. Louis, MO, USA). Resistant cell lines were continuously cultured in the presence of 6 μM sorafenib (Santa Cruz Biotechnology, Dallas, TX, USA) to preserve drug resistance. CoCl_2_ (Panreac AppliChem, Barcelona, Spain) was added at 100 μM to mimic hypoxia. We used 300 μM CHX and 30 μM MG132 (Tocris Bioscience, Bristol, UK) as protein synthesis and proteasome inhibitors, respectively.

Besides, we inhibited histone deacetylation using 10, 50, and 100 nM of the HDACs inhibitor TSA (AdooQ^®^ Bioscience, Irvine, CA, USA) and methylation with 10 and 100 μM of the DNMT inhibitor 5-Aza (MedChemExpress, Sollentuna, Sweden).

### 4.2. Growth Curve Based on Crystal Violet Staining

Treated cells were washed with ice-cold PBS and fixed with 4% paraformaldehyde (Thermo Fisher Scientific, Waltham, MA, USA) in PBS for 10 min. After fixing, cells were washed with Milli-Q water and stained with 0.1% crystal violet (Labkem, Spain) in 10% ethanol 20 min. After 3 washes, 10% acetic acid was added to each well followed by 20 min of incubation with shaking. It was taken 0.5–1 mL of dye and diluted 1:4 in water to subsequently measure absorbance at 590 nm spectral wavelength using a microtiter plate reader (Synergy™ HT Multi-Mode Microplate Reader; BioTek Instruments, Inc., Winooski, VT, USA).

### 4.3. Immunofluorescence and Laser Confocal Imaging

Cells were seeded on gelatin-coated coverslips. After treatments, cells were fixed with 4% paraformaldehyde (Thermo Fisher Scientific) for 15 min, washed with PBS, and permeabilized with 0.2% saponin (Sigma-Aldrich) and 1% fatty acid-free BSA (Sigma-Aldrich) in PBS, all at room temperature. After washing cells with PBS, they were stained overnight at 4 °C with Ki67 (sc-23900, Santa Cruz Biotechnology), HIF-1α (ab2185, Abcam, Cambridge, UK), HIF-2α (ab199, Abcam), or BNIP3 (sc-56167, Santa Cruz Biotechnology) antibodies. Then, cells were washed with PBS and incubated for 1 h at room temperature with Alexa 488-conjugated antimouse (Z25002, Molecular Probes, Eugene, OR, USA) or Alexa 647-conjugated antirabbit (Z25308, Molecular Probes) IgGs. Coverslips were washed with PBS and mounted on glass slides with fluorescent mounting medium Fluoroshield™ containing 4′6-diamidino-2-phenylindole (DAPI) (Sigma-Aldrich) to be visualized in a Zeiss LSM 800 confocal laser scanning microscope (Zeiss AG, Jena, Germany). Confocal images were analyzed with ZEN software (Zeiss AG, Jena, Germany). Fluorescence quantification was performed using ImageJ software (NIH, Bethesda, MD, USA) and the corrected total cell fluorescence (CTCF) formula.

### 4.4. Western Blot Assay

After treatments, cultured cells were washed with ice-cold PBS and lysed in a homogenization buffer containing 0.25 mM sucrose, 10 mM Tris, and 1 mM EDTA with protease and phosphatase inhibitors (Roche Diagnostics, Basel, Switzerland) by sonication during two pulses of 20 s at 60% amplitude, and centrifuged at 14,000 g for 10 min. Equal amounts of protein were separated by SDS-PAGE and transferred to PVDF membranes (Bio-Rad, Hercules, CA, USA) [62]. Membranes were blocked for 1 h at room temperature using 5% milk powder in a PBS solution with Tween 20 (Sigma-Aldrich) at 0.05% (PBS-T) and incubated overnight at 4 °C with the following primary antibodies: HIF-1α (ab2185, Abcam), HIF-2α (ab199, Abcam), β-actin (A3854, Sigma-Aldrich), Bax (sc-493, Santa Cruz Biotechnology), cleaved caspase-3 (#9661, Cell Signaling, Beverly, MA, USA), and BNIP3 (sc-56167, Santa Cruz Biotechnology). After three PBS-T washes, membranes were incubated for 1 h at room temperature with antirabbit (31460, Thermo Fisher Scientific) or antimouse (P0260, Dako, Glostrup, Denmark) HRP-conjugated secondary antibodies. Proteins were visualized using Pierce ECL western blotting substrate (Thermo Fisher Scientific). Band density was quantified employing ImageJ software (NIH).

We found considerable differences in β-actin levels between HepG2 and the resistant cells. Therefore, we used PCNA (sc-56, Santa Cruz Biotechnology) as housekeeping, since we could not employ a cytoskeletal protein. PCNA validity as an internal reference gene has been reported in several studies from different species, including human, determining its stability [63,64,65].

### 4.5. Microarray and Gene Expression Analysis

After 72 h, cells were harvested with TRIzol^®^ Reagent (Invitrogen, Merelbeke, Belgium) and RNA was isolated with the RNeasy Kit (Qiagen, Chatsworth, CA, USA) according to the manufacturer’s instructions. RNA quality was assessed with the Agilent 2100 BioAnalyzer (Agilent, Palo Alto, CA, USA). Affymetrix Human Gene 1.0 ST Array (Affymetrix) was used as platform. Microarray data was analyzed with the Limma package from Bioconductor (http://www.bioconductor.org) [19]. Data were analyzed using Webgestalt2013 bioinformatic suit (http://www.webgestalt.org/webgestalt_2013/). Data are available at NCBI, GEO series GSE62813.

### 4.6. Flow Cytometry of SubG1 Cell Population

Cells were harvested via trypsinization 48 h after treatments, being then collected by centrifuging at 350 *g* for 5 min, washed with ice-cold PBS, and they were again centrifuged at the same conditions. Approximately 1 × 10^6^ cells per sample were fixed with 70% ethanol in PBS for 2 h at 4 °C and, after this interval, centrifuged at 850 *g* for 5 min. Then, cells were washed again, centrifuged at these conditions, and incubated with 0.5 mL PI/RNase Staining Buffer (BD Pharmingen™, Franklin Lakes, NJ, USA) for 15 min at room temperature in dark conditions. Using red propidium-DNA fluorescence, 5000 events were acquired for each sample with a FACSCalibur Flow Cytometer (Becton Dickinson, San José, CA, USA) and the CellQuest software. The percentage of cells in subG1 phase was determined by Weasel analytical software (WEHI, Melbourne, VIC, Australia).

### 4.7. Gene Silencing

Commercial siRNAs against EPAS-1 (i.e., HIF-2α) (sc-35316), HIF-1α (sc-35561), BNIP3 (sc-37451), and control siRNA encoding a non-targeting sequence (sc-36869) were purchased from Santa Cruz Biotechnology, since this provider supplies each siRNA as a pool of different target specific 19-25 nt siRNAs designed to knockdown expression of gene of interest. The siRNAs were introduced into cells by reverse transfection using Lipofectamine^®^ RNAiMAX Reagent (Thermo Fisher Scientific) according to the manufacturer’s protocol. Five hours after transfection, media were replaced for complete DMEM-high glucose and, after 24 h, cells were treated to finally be subjected to viability and Western blot assays.

### 4.8. Cell Viability Assay

After treatments, media were removed and PBS washing was made followed by addition of a 1:10 free-serum medium solution of 3-(4,5-dimethylthiazol-2-yl)-2,5-diphenyl-tertazolium bromide (MTT) (Sigma-Aldrich) dissolved in PBS at 5 mg/mL for 3 h at 37 °C. Then, MTT-containing media were replaced by DMSO to dissolve MTT precipitates. The optical densities were measured at 560 nm spectral wavelength using the microtiter plate reader (Synergy™ HT Multi-Mode Microplate Reader; Bio-Tek Instruments, Inc.).

### 4.9. Real-Time (q) Reverse Transcriptase (RT)-Polymerase Chain Reaction (qRT-PCR), and RT-PCR

After treatments, total RNA was isolated using TRIzol^®^ Reagent (Applied Biosystems, Carlsbad, CA, USA) according to the manufacturer’s instructions. Residual DNA was removed using RQ1 RNase-free DNase kit (Promega, Madison, WI, USA) and subsequently, total RNA (500 ng) was reverse transcribed to cDNA using a high capacity cDNA reverse transcription kit (Applied Biosystems). For qRT-PCR experiments, cDNA was amplified using *Power* SYBR™ Green PCR Master Mix (Applied Biosystems) on the StepOnePlus Real-Time PCR System (Applied Biosystems). Relative changes in gene expression levels were determined using the 2^−ΔΔCt^ method [66]. For RT-PCR experiments, cDNA was amplified using the KAPA HiFi HotStart ReadyMix PCR Kit (Kapa Biosystems, Wilmington, MA, USA) on the MJ Research PTC-200 Thermal Cycler (Marshall Scientific, Hampton, NH, USA) following these conditions: preliminary denaturation at 95 °C for 5 min; 30 cycles of denaturation at 94 °C for 30 s, annealing at 60 °C for 45 s, and elongation at 72 °C for 30 s; and a final elongation step at 72 °C for 10 min. The RT-PCR products were loaded onto 2% agarose gels and visualized with GelRed^®^ Nuclei Acid Gel Stain (41003, Biotium, Fremont, CA, USA) using the ChemiDoc™ XRS Universal Hood II and the Quantity One^®^ software (Bio-Rad). Human primers employed in qRT-PCR and RT-PCR assays were as follows: BNIP3 forward 5′-CGCAGACACCACAAGATACC-3′ and reverse 5′-TCTTCATGACGCTCGTGTTC -3′; 18S rRNA forward 5′-GGCGCCCCCTCGATGCTCTTAG-3′ and reverse 5′-GCTCGGGCCTGCTTTGAACACTCT-3′. 18S rRNA gene was used as an internal control.

### 4.10. Methylation-Specific PCR (MSP)

After treatments, total DNA was extracted using Phenol:Chloroform:Isoamyl Alcohol 25:24:1 Saturated with 10 mM Tris, pH 8.0, 1 mM EDTA (Sigma-Aldrich) following the manufacturer’s instructions. DNA was modified using the bisulfite conversion EZ DNA Methylation™ Kit (Zymo Research, Irvine, CA, USA) following the manufacturer’s protocol. Methylation-specific PCR (MSP) was carried out employing specific primers to detect methylated (M) and unmethylated (U) DNA: M-BNIP3 forward 5′-TAGGATTCGTTTCGCGTACG-3′ and reverse 5′-ACCGCGTCGCCCATTAACCGCG-3′; U-BNIP3 forward 5′-TAGGATTTGTTTTGTGTATG-3′ and reverse 5′-ACCACATCACCCATTAACCACA-3′. Amplification was performed in the MJ Research PTC-200 Thermal Cycler (Marshall Scientific) and the MSP conditions were as follows: initial denaturation at 95 °C for 15 min, 35 cycles of denaturation at 94 °C for 30 s, reassociation at 58 °C for 50 s, extension at 72 °C for 1 min, and final extension at 72 °C for 10 min. The MSP products were loaded onto 2% agarose gels and visualized with GelRed^®^ Nuclei Acid Gel Stain (41003, Biotium) using the ChemiDoc™ XRS Universal Hood II and the Quantity One^®^ software (Bio-Rad).

### 4.11. Statistical Analysis

Differentially expressed genes from microarray were assessed using a moderated t-test. Resulting p-values were corrected for multiple testing with Benjamini-Hochberg to control false discovery rate [19]. For selecting differentially expressed genes, a cut-off of Δlog (^2^log FC) > +0.7 or < −0.7 and a corrected *p* < 0.05 was applied. All other results were expressed as mean values ± SD of three independent experiments. They were analyzed by the statistical package GraphPad Prism 6 (San Diego, CA, USA) using unpaired t-test or one-way, two-way ANOVA followed by Tukey post-hoc test to measure differences between the different groups, considered statistically significant when *p* < 0.05.

## 5. Conclusions

In summary, HIFs stabilization, even under normoxia, triggers overactivation of adaptive cell response to hypoxic microenvironment and seems to be responsible at least in part for loss of sensitivity to sorafenib of HCC cells. Furthermore, promoter hypermethylation causes BNIP3 knockdown, evading BNIP3-mediated cell death under hypoxia. Both mechanisms, represented in the graphical abstract, seem to constitute central hallmarks of sorafenib resistance acquisition. Therefore, avoiding HIFs stabilization and the subsequent prosurvival signaling, as well as BNIP3 expression reactivation, could be a promising therapeutic strategy to overcome sorafenib resistance in advanced HCC.

## Figures and Tables

**Figure 1 cancers-11-01984-f001:**
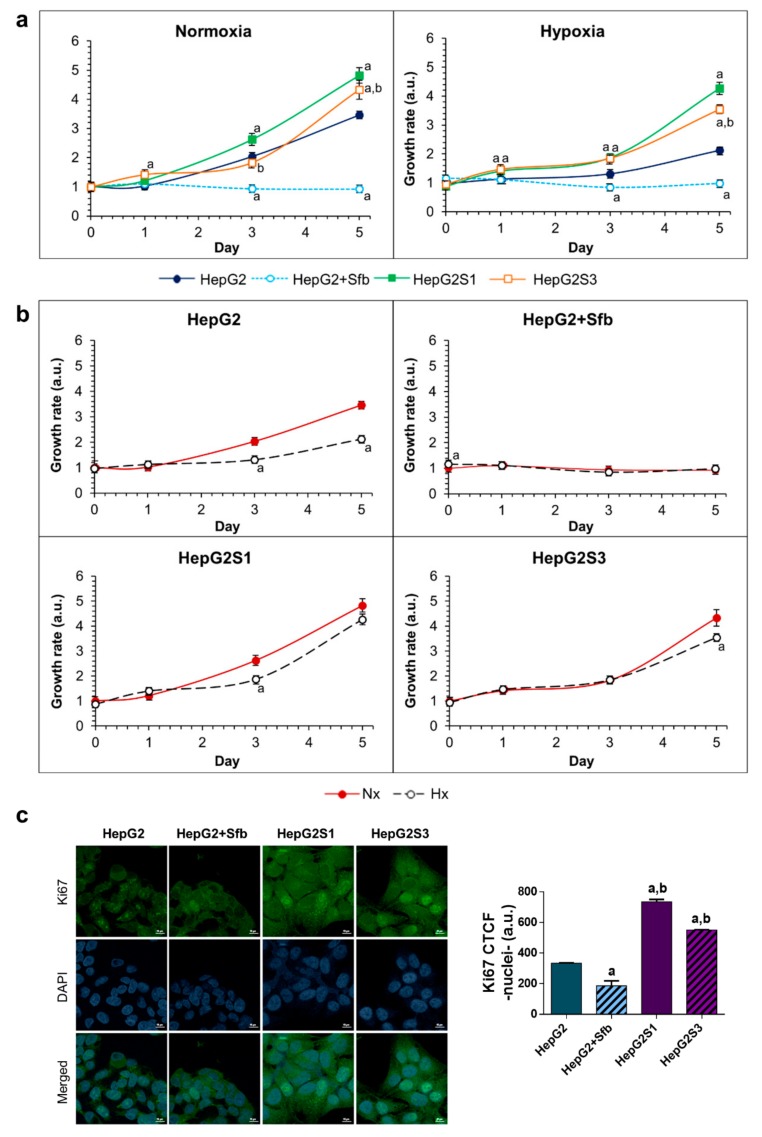
Growth dynamics and cell proliferation: (**a**) Comparison of cell growth between HepG2S1 and HepG2S3 sorafenib resistant lines and HepG2 parental line under normoxia and hypoxia. ^a^
*p* < 0.05 vs. non-treated HepG2 cells, ^b^
*p* < 0.05 significant differences between sorafenib resistant cells; (**b**) Comparison of cell growth between normoxia and hypoxia within the same cell line. ^a^
*p* < 0.05 vs. normoxic cells; (**c**) Comparison of cell proliferation between resistant cell lines and HepG2 cells after 24 h incubation under hypoxia. Confocal images of Ki67 immunofluorescence staining (green) show Ki67 expression. 4′,6-diamidino-2-phenylindole (DAPI) staining (blue) denotes cell nucleus. Magnification: 63X, scale bar: 10 µm. ^a^
*p* < 0.05 and ^b^
*p* < 0.05 vs. non-treated and sorafenib-treated HepG2 cells, respectively. Data from (**a**–**c**) are expressed as mean values of arbitrary units (a.u.) ± SD of three independent experiments.

**Figure 2 cancers-11-01984-f002:**
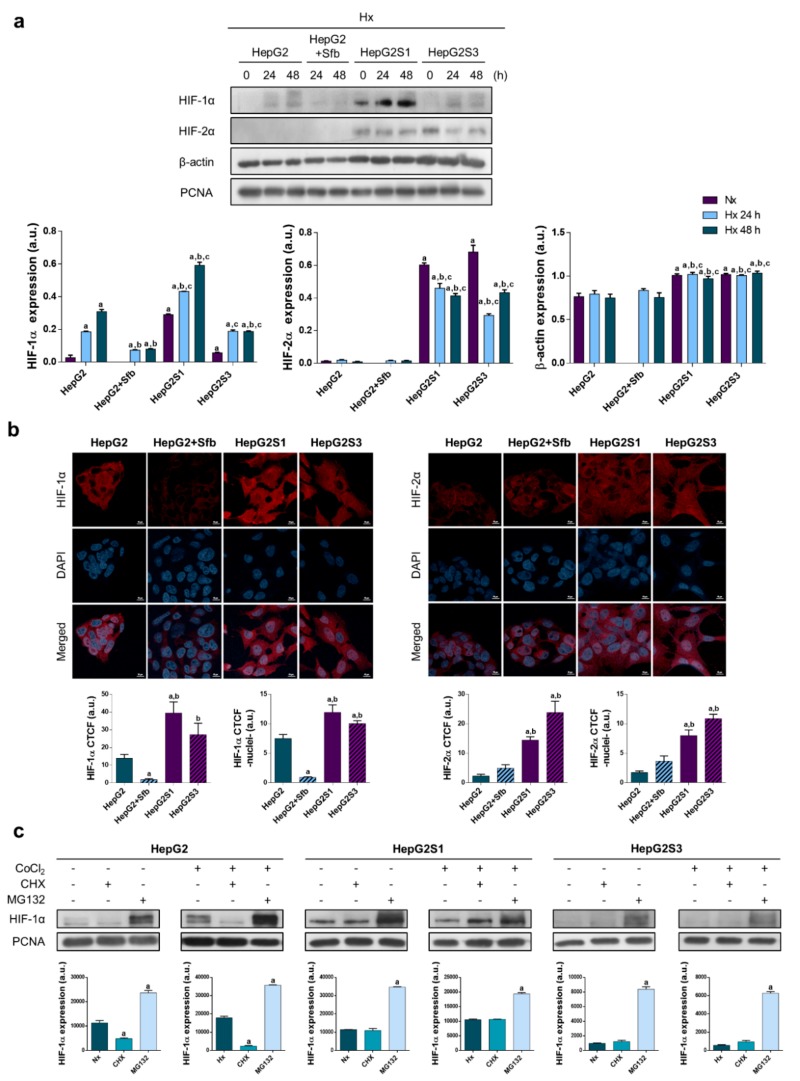
Cell modulation of hypoxia response in sorafenib resistance: (**a**) Effect of hypoxia on protein expression. Lanes 0 h show normoxic basal protein levels. ^a^
*p* < 0.05 vs. normoxic non-treated HepG2 cells, ^b^
*p* < 0.05 and ^c^
*p* < 0.05 vs. hypoxic non-treated and sorafenib-treated HepG2 cells, respectively, at each time point; (**b**) Confocal images of hypoxia-inducible factor (HIF)-1α (left panel) and HIF-2α (right panel) immunofluorescence staining (red) show HIFs expression after incubation under hypoxia for 24 h. DAPI staining (blue) denotes cell nucleus. Magnification: 63×, scale bar: 10 µm. Bar graphs at left position represent total expression whereas bar graphs at right position represent nuclear translocation for each marker analyzed. ^a^
*p* < 0.05 and ^b^
*p* < 0.05 vs. non-treated and sorafenib-treated HepG2 cells, respectively; (**c**) Evaluation of HIF-1α protein synthesis and degradation processes. ^a^
*p* < 0.05 vs. normoxia/hypoxia within the same cell type. Data from (**a**–**c**) are expressed as mean values of arbitrary units (a.u.) ± SD of three independent experiment. Full-length immunoblots are presented in Appendix A.

**Figure 3 cancers-11-01984-f003:**
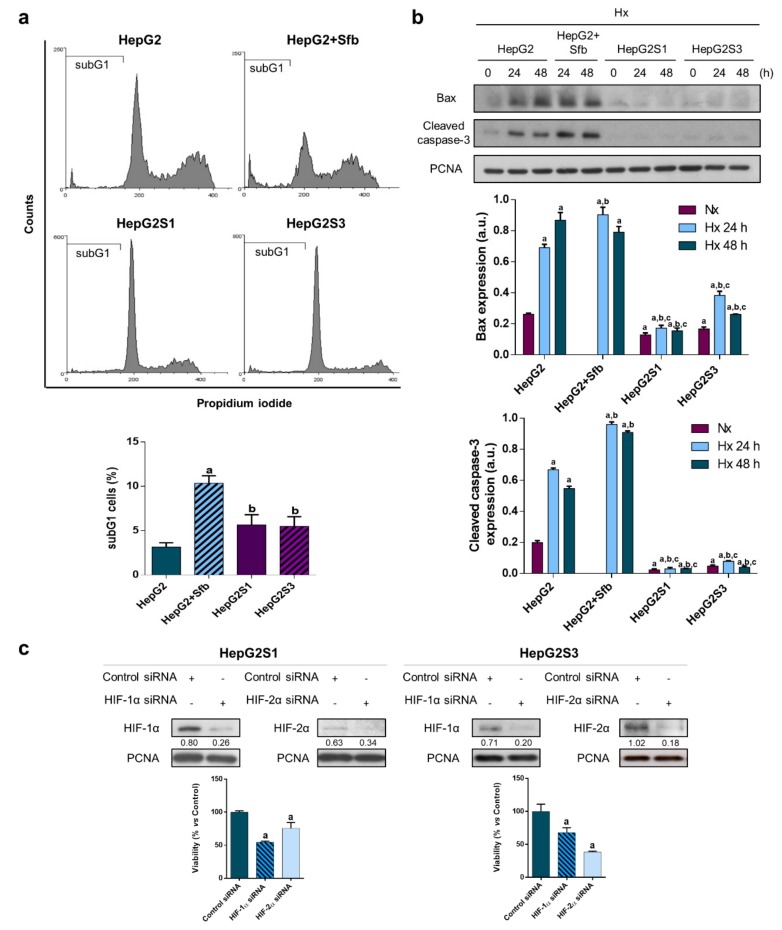
Cell death assessment and involvement of HIFs in sorafenib resistant cells survival: (**a**) SubG1 cell population evaluation after incubation under hypoxia for 48 h. ^a^
*p* < 0.05 significant differences between sorafenib-treated and non-treated HepG2 cells, ^b^
*p* < 0.05 significant differences between resistant cells and sorafenib-treated HepG2 cells; (**b**) Analysis of Bax and cleaved caspase-3 protein expression. Lanes 0 h show normoxic basal protein levels. ^a^
*p* < 0.05 vs. normoxic non-treated HepG2 cells, ^b^
*p* < 0.05 and ^c^
*p* < 0.05 vs. hypoxic non-treated and sorafenib-treated HepG2 cells, respectively, at each time point; (**c**) HIFs silencing: representative immunoblots and cell viability analysis after 24 h under hypoxia. Densitometry reading of each band (relative to its correspondent proliferation cell nuclear antigen (PCNA) band) is shown under the immunoblots. ^a^
*p* < 0.05 vs. control small interfering RNA (siRNA) cells. Data from (**a**) and (**c**) graphics are expressed as a percentage of mean values ± SD of experiments performed in triplicate. Data from (**b**) are expressed as mean values of arbitrary units (a.u.) ± SD of three independent experiments. Full-length immunoblots are presented in Appendix A.

**Figure 4 cancers-11-01984-f004:**
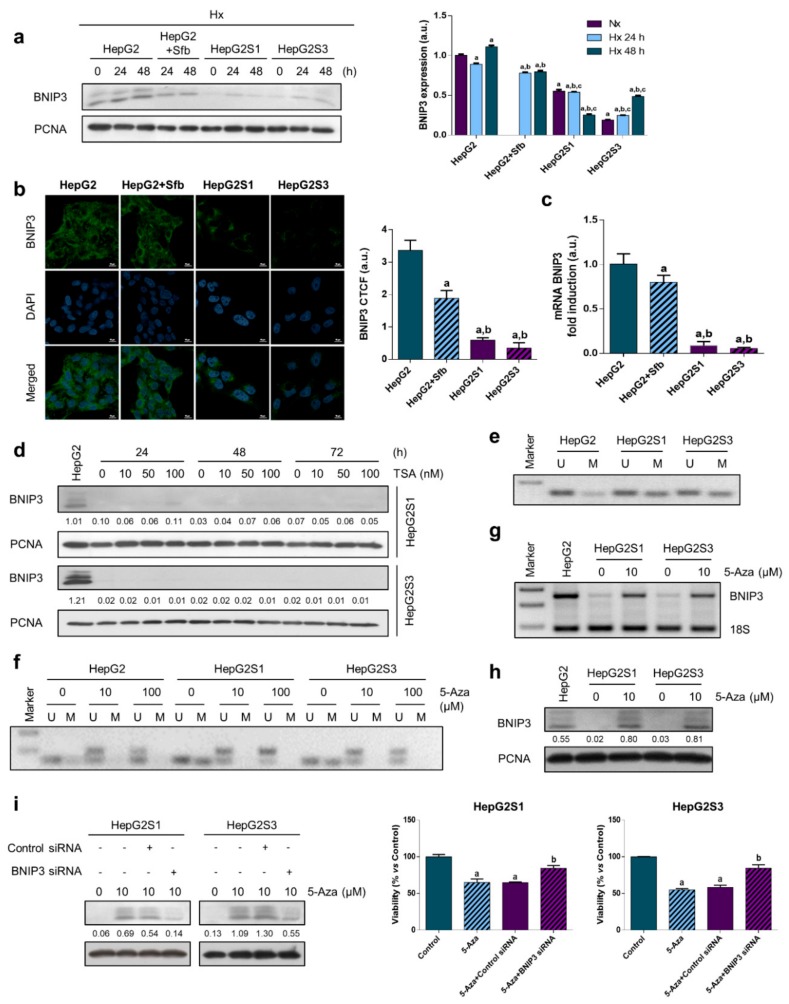
Analysis of Bcl-2 interacting protein 3 (BNIP3) expression under hypoxic conditions and role of epigenetic modulations on chemoresistance-associated BNIP3 downregulation: (**a**) BNIP3 protein levels. Lanes 0 h show normoxic basal protein levels. ^a^
*p* < 0.05 vs. normoxic non-treated HepG2 cells, ^b^
*p* < 0.05 and ^c^
*p* < 0.05 vs. hypoxic non-treated and sorafenib-treated HepG2 cells, respectively, at each time point; (**b**) Confocal images of BNIP3 immunofluorescence staining (green) show BNIP3 expression after incubation under hypoxia for 24 h. DAPI staining (blue) denotes cell nucleus. Magnification: 63X, scale bar: 10 µm. ^a^
*p* < 0.05 and ^b^
*p* < 0.05 vs. non-treated and sorafenib-treated HepG2 cells, respectively; (**c**) BNIP3 mRNA levels 24 h under hypoxia were measured by qRT-PCR. ^a^
*p* < 0.05 and ^b^
*p* < 0.05 vs. non-treated and sorafenib-treated HepG2 cells, respectively; (**d**) Effect of inhibiting histone deacetylases (HDACs) on BNIP3 expression by trichostatin-A (TSA) addition. HepG2S1 and HepG2S3 cells were incubated under hypoxia during the last 24 h of every TSA treatment. First lane shows standard BNIP3 protein levels from HepG2 cells after 24 h incubation under hypoxia. Densitometry reading of each band (relative to its correspondent PCNA band) is shown under the immunoblots; (**e**) Methylation status of the BNIP3 promoter was examined performing methylation-specific PCR (MSP) after 24 h of hypoxia. Specific BNIP3 primers were used to amplify either unmethylated (U) or methylated (M) DNA; (**f**) Effect of 5-aza-2′-deoxycytidine (5-Aza) treatment on the methylation status of the BNIP3 promoter. Cells were incubated with the demethylation agent for 48 h before exposure to hypoxia plus 5-Aza for further 24 h; Impact of 5-Aza treatment during 72 h (last 24 h also under hypoxia) on (**g**) BNIP3 mRNA and (**h**) protein levels. HepG2 lanes show basal mRNA and protein levels after 24 h under hypoxia, respectively; (**i**) Effect of demethylation by 5-Aza alone or in conjunction with BNIP3 silencing on the BNIP3 expression and the resistant cell viability under hypoxia. ^a^
*p* < 0.05 significant differences between 5-Aza-treated and non-treated cells, ^b^
*p* < 0.05 significant differences between 5-Aza-treated BNIP3 siRNA and 5-Aza-treated control siRNA cells. Densitometry reading of each band (relative to its correspondent PCNA band) from (**h**) and (**i**) is shown under the immunoblot. Data from (**a**–**c**) are expressed as mean values of arbitrary units (a.u.) ± SD of three independent experiments. Data from (**i**) are expressed as a percentage of mean values ± SD. Full-length immunoblots are presented in Appendix A.

**Table 1 cancers-11-01984-t001:** Relative expression of RNA for apoptosis markers in sorafenib resistant HepG2S1 cells vs. parental HepG2 cells under normoxia determined by microarray *.

Gene Symbol	Full Name	Regulation	^2^logFC HepG2S1/HepG2	Corrected *p*
BAX	BCL2 associated X apoptosis regulator	Down	−0.93	5.74 × 10^−9^
BCL2	BCL2 apoptosis regulator	Up	+1.25	1.43 × 10^−8^
BCL2L1	BCL2 like 1	Down	−1.21	1.59 × 10^−10^
BIRC3	Baculoviral IAP repeat containing 3	Up	+5.06	1.32 × 10^−15^
CASP2	Caspase 2	Up	+0.95	1.18 × 10^−8^
CASP3	Caspase 3	Down	−2.06	4.59 × 10^−11^
CASP8	Caspase 8	Down	−0.81	7.98 × 10^−9^
CASP9	Caspase 9	Up	+1.21	1.76 × 10^−9^
CASP10	Caspase 10	Down	−1.23	6.76 × 10^−8^
HRK	Harakiri BCL2 interacting protein	Up	+1.34	1.17 × 10^−7^
PMAIP1	Phorbol-12-myristate-13-acetate-induced protein 1	Up	+2.45	9.17 × 10^−13^
PTPN13	Protein tyrosine phosphatase non-receptor type 13	Up	+1.97	5.59 × 10^−10^
TNFRSF10B	TNF receptor superfamily member 10b	Down	−1.23	5.04 × 10^−11^

* Microarray data derive from three independent experiments.

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
