# Peer review of "Stabilization of Hypoxia-Inducible Factors and BNIP3 Promoter Methylation Contribute to Acquired Sorafenib Resistance in Human Hepatocarcinoma Cells"

_cancers, 2019, doi:10.3390/cancers11121984_

Round 1

Reviewer 1 Report

The manuscript titled "Stabilization of Hypoxia-Inducible Factors and BNIP3 Promoter Methylation Contribute to Acquired Sorafenib Resistance in Human Hepatocarcinoma Cells" by Méndez-Blanco et al., aims to study the mechanism of sorafenib resistance and the role of HIF-1a and HIF-1b. The authors also show the role of Bax as well the antipapoptotic IAP.  The study is well-organized, the results are solid and the discussion is supported by the data. 

Author Response

We greatly appreciate your comments and your contribution as a reviewer. We hope that the impact of this article is adequate and useful for the scientific community.

Reviewer 2 Report

In this article Méndez-Blanco et al demonstrate that HIF and BNIP3 promoter methylation are important for sorefanib resistance in HCC. For this demonstration they used hypoxia conditions and showed Figure 1 that growth rates of their cells were different in normoxia and hypoxia conditions. These results are further exemplified by ki67 staining.

Nevertheless, they used CoCL2 to mimic hypoxia.  CoCL2 is an inducer of HIF1alpha but does not recapitulate all the effects of hypoxia. The authors have to precise in the abstract that they used CoCL2 to mimic hypoxia. Moreover Hypoxia chambers is the gold standard method and the authors must use this equipment and not CoCL2 to publish in good journal such Cancers.

Then they showed that Hypoxia induced HIF1 and/or 2 alpha depending of the cell lines.

As they observed Beta actin rates were not constant and they showed that PCNA did not change. Based on these results they decided to choose PCNA as a control. Nevertheless, PCNA is also a cell cycle marker: ie expression of PCNA is well known to vary among cell cycle phases. So the authors showed that cell proliferation was affect but not PCNA. How the authors explain this discrepancy? The authors have to use other control such as gapdh, tubulin…

Western blots were quantified so the authors have to provide original photographs of all western blots in supplemental data.

In Figure 3 the authors showed Sub-G1, it’s an accumulation of degraded “nuclei” and not apoptotic nuclei so line 207 is not corrected to claim that cells are apoptotic. If it’s true please provided TUNEL or PS translocation analyses.

In Fig3b please provide better western blot analyses for bax and cleaved caspase 3, first line is not correctly load and to avoid such a smile in western blot it will be fine to adapt western blot condition.

Author Response

We agree with the referee that the use of hypoxic chambers is an ideal method to induce hypoxia. However, CoCl2 is a valid alternative (see Wu et al., J. Vis. Exp., 2011, PMID 21860378) widely used to mimic hypoxia by inducing HIFs. Since 2003 more than 350 articles have employed this method to induce HIFs stabilization (37 in 2019) and results using CoCl2 as hypoximimetic have been published in journals as prestigious as Sci. Rep. (IF 4.01), FASEB J. (IF 5.40), J. Cell. Mol. Med. (IF 4.67), J. Biol. Chem. (IF 4.11), Transplantation (IF 4.74), Cell Death. Dis. (IF 6.00) or Nat. Microbiol. (IF 14.63). Moreover, hypoxia chambers conditions for different cell lines have to be optimized in pressure and time and they do not have to be the same.

Our sorafenib resistant cell model does not allow us to use cytoskeletal proteins as loading control because of the marked variation in the morphology of resistant cells in comparison to the parental line HepG2. This change in cell morphology is a direct consequence of the process of acquisition of resistance to sorafenib and the reason why actin and tubulin are not valid housekeepers for us. In addition, our work is performed under hypoxic conditions and GAPDH is a well-recognized target gene of HIF-1. This fact was already reported by the 2019 Nobel Prize Gregg L. Semenza (Nat. Rev. Cancer, 2003, PMID 13130303) and consequently makes also not possible to employ GAPDH as loading control in our Western blot assays.

PCNA is a nuclear protein which acts as a cofactor of DNA polymerase delta. It is true that PCNA is associated with cellular division and DNA replication, but this precisely is the reason by which its expression is stable in proliferating cells (e.g. our HCC cells) and its levels do not vary with cell cycle status in mammalian cells. Therefore, PCNA represents a valid housekeeper to our Western blot experiments because our cell populations undergo proliferation. This information is supported by several Western blots guides from different suppliers that include PCNA as a good option as nuclear loading control. See the following web pages:

https://www.labome.com/method/Loading-Controls-for-Western-Blots.html

https://www.ptglab.com/news/blog/loading-control-antibodies-for-western-blotting/

https://www.cusabio.com/c-20864.html

https://www.abcam.com/primary-antibodies/loading-control-guide

Furthermore, several research articles found in scientific literature support PCNA as internal reference gene. Some of them are already included among the references of our manuscript (see 62, 63 and 64). Based on the explanations and supports reported here, we respectfully disagree with the reviewer and consider PCNA as a valid loading control in our work.

We clearly agree with the referee and original photographs of all Western blots are already presented as Supplementary Materials (see Supplementary Figure S1).

We agree with the referee comment that line 207 is not totally correct given that evaluation of subG1 phase provides information about cell death rather than apoptosis. Accordingly, all sentences in the text where accumulation of subG1 cells was related to apoptosis instead of cell death in general, including line 207, were properly modified in the revised manuscript.

Apart from assessing subG1 phase cell percentage, we completed the study of cell death with the microarray analysis of the KEGG-pathway for apoptosis (hsa04210) and the evaluation of the expression of the apoptotic markers Bax and cleaved caspase-3 by Western blot. Therefore, although TUNEL or PS translocation analyses could be useful, we estimate that the cell death and apoptosis data already shown in the manuscript are representative and adequate.

Analysis of Bax and cleaved caspase-3 expression was repeated after the correct adaption of Western blot conditions in order to improve the quality of the images (see new Figure 3b).

Reviewer 3 Report

In the title, the term "stabilization" is vague and not easy to understand. Because title should be most attractive first message to readers, it should be more specific, like "downregulation".

Author Response

The term “stabilization” is extensively used in the scientific literature to refer specifically to the restriction of prolyl hydroxylases (PHDs) activity, which consequently avoids the proteasomal degradation of hypoxia-inducible factors (HIFs) allowing the HIF protein to accumulate and induce the transcription of its target genes (i.e. Schley et al., Am. J. Pathol., 2012, PMID 22944601; Maroni et al., Exp. Cell Res., 2015, PMID 25447306; Curtis et al., FASEB J., 2015, PMID 25326537; Hoppe et al., Proc. Natl. Acad. Sci. USA, 2016, PMID 27091985; Khan et al., Neural Regen. Res., 2017, PMID 28616019; Fortenbery et al., Cell. Mol. Biol. Lett., 2018, PMID 30305827; Müller et al., Sci. Rep., 2018, PMID 29934555; Villacampa et al., Angiogenesis, 2019, PMID 31583505; Kurokawa et al., PLoS One, 2019, PMID 31513628; Anton et al., Front. Physiol., 2019, PMID 31263422; and many others). This is now better clarified in the text of the revised manuscript. For these reasons, we respectfully disagree with the referee suggestion and we consider the term “stabilization” as the most appropriate.

Reviewer 4 Report

Study by Carolina Méndez-Blanco et al. "Stabilization of Hypoxia-Inducible Factors and BNIP3 PromoterMethylation Contribute to Acquired Sorafenib Resistance in HumanHepatocarcinoma Cells" is good and informative, however it lacks proper controls to justify conclusions. My major comments are as follows:

1. Authors should include more HCC cell lines (such as Hep3B, Huh1/7, HLE, HLF1-4, etc.) to confirm their key findings.    2. Authors should show that BNIP3 over-expression can rescue the drug resistance in resistant cell lines, as 5-AZA is Pan-DNMT inhibitor and effects global DNA methylation.    3. How HIF1a regulated BNIP3 methylation during drug resistance, or it is merely an associated event is still not clear.

Author Response

We agree that performing all the analysis employing more HCC cell lines would contribute to highlight the results obtained here. However, the additional cell lines that could be included in our study could not be well-known HCC cell lines as the Hep3B, Huh1/7 lines. Given that we aim to elucidate mechanisms of sorafenib resistance, it would only be adequate to confirm the results from our HepG2-derived sorafenib resistant model using other sorafenib resistant cell lines. Nonetheless, there is an important time limitation due to the fact that, as we indicated in the methodology section (see reference 19), it takes several months to develop and characterize each sorafenib resistant cell line. Additionally, we have considered that many of the available cell lines have integrated viral sequences like HBV (Hep3B) and have alterations in p53 (mutations or truncated). HepG2 is a cell line that strongly mimics untransformed human liver cells in behavior (receptor expression and response to stimuli) and is widely used to study the liver protein expression.

We agree with the referee that 5-Aza reverses global DNA methylation; hence, testing the real implication of BNIP3 overexpression in the sorafenib resistance is a good idea. Because we have no material and experience enough to overexpress BNIP3 in two weeks, we decided to do an alternative experiment. In order to demonstrate that the reactivation of BNIP3 expression is responsible for the rescue of the sorafenib resistance in our in vitro model, we assessed the effect on cell viability of the treatment with the DNMT inhibitor 5-Aza in conjunction with BNIP3 silencing. Results showed that BNIP3 silencing in combination with 5-Aza administration significantly increases the cell viability which had been reduced after 5-Aza individual treatment. These findings prove that the decrease of viability observed in the 5-Aza non-silenced cells in comparison to the non-treated non-silenced cells is due to the reactivation of BNIP3 expression, at least in part, which reinforces the relevant role displayed by the BNIP3 epigenetic silencing in the survival ability of our sorafenib resistant lines (see new Figure 4i).

HIF-1 alpha is a transcription factor that has been widely related to the regulation of BNIP3 transcription through its binding to the hypoxia-response element (HRE) located in BNIP3 promoter. According to your comment, few studies have found a possible relationship between hypoxic microenvironment and promoter methylation (i.e. Zhang et al., Cell Physiol. Biochem., 2018, PMID 29414807; Yoon et al., Cell Biochem. Funct., 2017, PMID 29082591), but only Zhang et al. related it to HIF-1 and none of them analyzed BNIP3 promoter specifically. Furthermore, several researches have reported BNIP3 promoter methylation, but they have not linked this epigenetic regulation to HIF-1. In these studies, the unique finding was the physical impossibility of HIF-1 to bind to the BNIP3 promoter because of the methylation (see references 18 and 57). Moreover, An et al. suggested that BNIP3 methylation results from the DNMT1 induction dependent of the MAPK pathway in pancreatic cancer cells (see reference 15); nevertheless, HIF-1 alpha has not been associated to the regulation of BNIP3 promoter methylation. Altogether, the association between HIF-1 and BNIP3 promoter methylation remains still unclear although it could be an interesting issue to investigate.

Round 2

Reviewer 2 Report

none

Reviewer 4 Report

The study by Carolina et al. has improved and addressed the questions. The feel the study has importance to the field and translational value.

Sincerely,

Brijesh